# A mechanism for the activation of the mechanosensitive Piezo1 channel by the small molecule Yoda1

Wesley M. Botello-Smith[1], Wenjuan Jiang [1], Han Zhang[1], Alper D. Ozkan[2], Yi-Chun Lin[1], Christine N. Pham[1], Jérôme J. Lacroix [2]* & Yun Luo [1]*

Mechanosensitive Piezo1 and Piezo2 channels transduce various forms of mechanical forces into cellular signals that play vital roles in many important biological processes in vertebrate organisms. Besides mechanical forces, Piezo1 is selectively activated by micromolar concentrations of the small molecule Yoda1 through an unknown mechanism. Here, using a combination of all-atom molecular dynamics simulations, calcium imaging and electrophysiology, we identify an allosteric Yoda1 binding pocket located in the putative mechanosensory domain, approximately 40 Å away from the central pore. Our simulations further indicate that the presence of the agonist correlates with increased tension-induced motions of the Yoda1-bound subunit. Our results suggest a model wherein Yoda1 acts as a molecular wedge, facilitating force-induced conformational changes, effectively lowering the channel's mechanical threshold for activation. The identification of an allosteric agonist binding site in Piezo1 channels will pave the way for the rational design of future Piezo modulators with clinical value.

---

[1] College of Pharmacy, Western University of Health Sciences, 309 E. Second St, Pomona, CA 91766, USA. [2] Graduate College of Biomedical Sciences, Western University of Health Sciences, 309 E. Second St, Pomona, CA 91766, USA. *email: jlacroix@westernu.edu; luoy@westernu.edu

Mechanosensitive Piezo channels form a small protein family encompassing only two paralogs in vertebrates, Piezo1 and Piezo2. In spite of their small family size, these channels undertake many important physiological functions in adult and developing animals, including osmotic homeostasis, visceral, and somatic mechanosensation, proprioception, blood flow sensing, and epithelial homeostasis[1–9]. Abnormal Piezo channel activity caused by inherited mutations, genetic manipulation or physiological regulation has been linked to a variety of pathological conditions such as xerocytosis, lymphedema, arthrogryposis, abnormal vascular development, sleep apnea, allodynia and the loss of proprioception[10–20].

Piezo1, the best studied Piezo channel, opens upon direct physical deformations of the lipid bilayer, such as increased membrane tension, and thus obeys the so-called force-from-lipid paradigm established for several mechanosensitive ion channels[21–24]. Recent advances in cryo-electron microscopy (cryo-EM) have enabled the capture of mouse Piezo1 (mPZ1) in non-conducting states, which presumably correspond to the canonical closed conformation populated in absence of external mechanical force[25–27]. These structures reveal a propeller-like structure with a central pore surrounded by three large peripheral domains called arms, or blades. Each of these peripheral domains is formed by a succession of helical bundles called Piezo repeats, or transmembrane helical units, and are connected to long intracellular beams that extend toward the intracellular side of the pore. The arc-shaped arms are anticipated to mediate a local curvature, or inverted dome shape, to the lipid bilayer and to act as the channel mechanosensory domains[25,26,28,29]. Although the mechanism by which tension-mediated rearrangements of the arms open the channel gate is presently unknown, a recent study suggests the arms and beams may efficiently transfer mechanical forces from lipids to the pore via a lever mechanism[28].

Besides mechanical stimulation, several synthetic small molecule Piezo1-selective agonists have been identified using high-throughput screening. Among them, Yoda1 possesses the highest potency, with a half maximal effective concentration ($EC_{50}$) and binding affinity to the purified Piezo1 protein in the range of 10–50 μM[28,30]. Despite the identification of mutations that abolish Yoda1-mediated activation[25,28,31], the location of the Yoda1 binding site and the mechanism by which Yoda1 modulates Piezo1 remain unknown. In principle, the Yoda1 binding site(s) could be identified by solving the structure of a Yoda1-Piezo1 complex, although the resolution of current cryo-EM structures (3.5 to 4 Å) may be too low for unambiguous identification of the small, 29-atoms Yoda1 molecule. Here, to identify this binding site, we perform microseconds-long all-atom molecular dynamics (MD) simulations of mPZ1 in the presence of multiple Yoda1 ligands and in absence or presence of membrane stretch. In this context, we have taken advantage of ANTON2 supercomputer, which has been used successfully to identify small molecule binding sites and binding pathways in membrane proteins[32–36]. We report the identification of an allosteric Yoda1 binding site and propose a simple mechanism by which Yoda1 may promote Piezo1 opening in the presence of membrane stretch.

## Results

### Piezo1 possesses a fenestrated pore and curves the membrane.
An all-atom reduced model of mPZ1 containing extracellular cap, central core, C-terminal domain (CTD), Piezo repeats A–C, and intracellular beam and clasp connecting repeat C and CTD, was built based on a high-resolution cryo-EM mPZ1 structure[26] and embedded in a POPC membrane in a box filled with water and 150 mM KCl (Supplementary Fig. 1). After equilibration, the

system was run on ANTON2 supercomputer with 2.0 femtosecond (fs) timestep for a total of 7.9 μs which includes in absence (4.8 μs), and then in presence of membrane tension (3.1 μs). The average root-mean-square deviation (RMSD) of the protein backbone atoms suggests that our protein-lipid system equilibrates after approximately 3.6 μs (Fig. 1a). This relatively long equilibration time is necessary to allow the lipid leaflets to form an inverted dome. We estimated the curvature radius of the POPC membrane bilayer spontaneously formed around the fully equilibrated Piezo1 to be 8–10 nm (see Methods and Fig. 1b, c). This radius value is very similar to the membrane radius of ~10 nm experimentally observed for mPZ1 reconstituted in liposomes[26]. Even though a large portion of peripheral arm region are missing, this result show that our reduced all-atom model of mPZ1 recaptures a realistic dome shape in a fully solvated membrane environment.

During the simulation, several hydrophobic residues (I2466, L2469, I2473, V2477, and F2480) remain pointing toward the pore lumen and occlude the putative ion permeation pathway during our simulation (0–4.8 μs, Fig. 1d). As anticipated from cryo-EM studies[25,26,29], a pore vestibule located below F2480 may be accessible from the cytosol via lateral fenestrations. Indeed, we observed high potassium ion density in this vestibule, while chloride ions were excluded from this region throughout our trajectory (Fig. 1e). Potassium ions occupy these fenestrations by interacting with several negatively charged glutamate residues, E2487, E2495, and E2496. This is consistent with the fact that charge neutralization of some of these residues abolish cation selectivity (Fig. 1f)[37]. Hence, our simulation supports that these lateral fenestrations may constitute the intracellular pathway for cations when the channel gate opens.

### Identification of a putative Yoda1 binding site.
20 Yoda1 ligands (L1-L20) were added to the simulation system to ensure effective sampling of potential binding sites (Fig. 2a and Supplementary Movie 1). A root-mean-square fluctuation (RMSF) analysis was used to determine the mobility of each molecule over time (Fig. 2b). The binding site corresponding to the most stable ligand, L13, is a narrow hydrophobic pocket located near residues 1961–2063 (Fig. 2c), which are critical for Yoda1-mediated Piezo1 activation[31]. Interestingly, L13 enters its binding pocket from the intracellular leaflet of the membrane (Fig. 2d), in agreement with excised patch-clamp recordings showing larger Yoda1-mediated currents upon application of Yoda1 on the intracellular side of the patch[28]. In this binding pocket, sandwiched between Piezo repeats A and B, three alanine point toward Yoda1: A1718, A2091 and A2094 (Fig. 2e). To validate this binding site, we sought to sterically disrupt or alter the interaction of Yoda1 with this pocket by individually inserting tryptophan residues at these positions. The three mutants were tested using calcium imaging in cells where the expression of endogenous Piezo1 is abolished (ΔPZ1)[38]. Calcium entry was monitored by co-transfecting cells with the calcium indicator GCaMP6m (GC6). Fluorescence analyses show that Yoda1-sensitivity is either totally abolished (A1718W), severely reduced (A2094W) or partially diminished (A2091W) in these mutants (Fig. 2f). The second most stable ligand from our RMSF plot, L20 (Fig. 2b), is located near several small residues (A1326, S1330, V1533, A1972, and A1973) (Fig. 2g). However, in sharp contrast to Trp mutants in the L13 binding site, individual replacement of small residues near L20 with tryptophan did not produce significant loss of Yoda1-sensitivity, ruling out this site as an effective Yoda1 agonist binding site (Fig. 2h).

### Functional characterization of Piezo1 mutants.
We next performed cell-attached pressure-clamp electrophysiology recordings

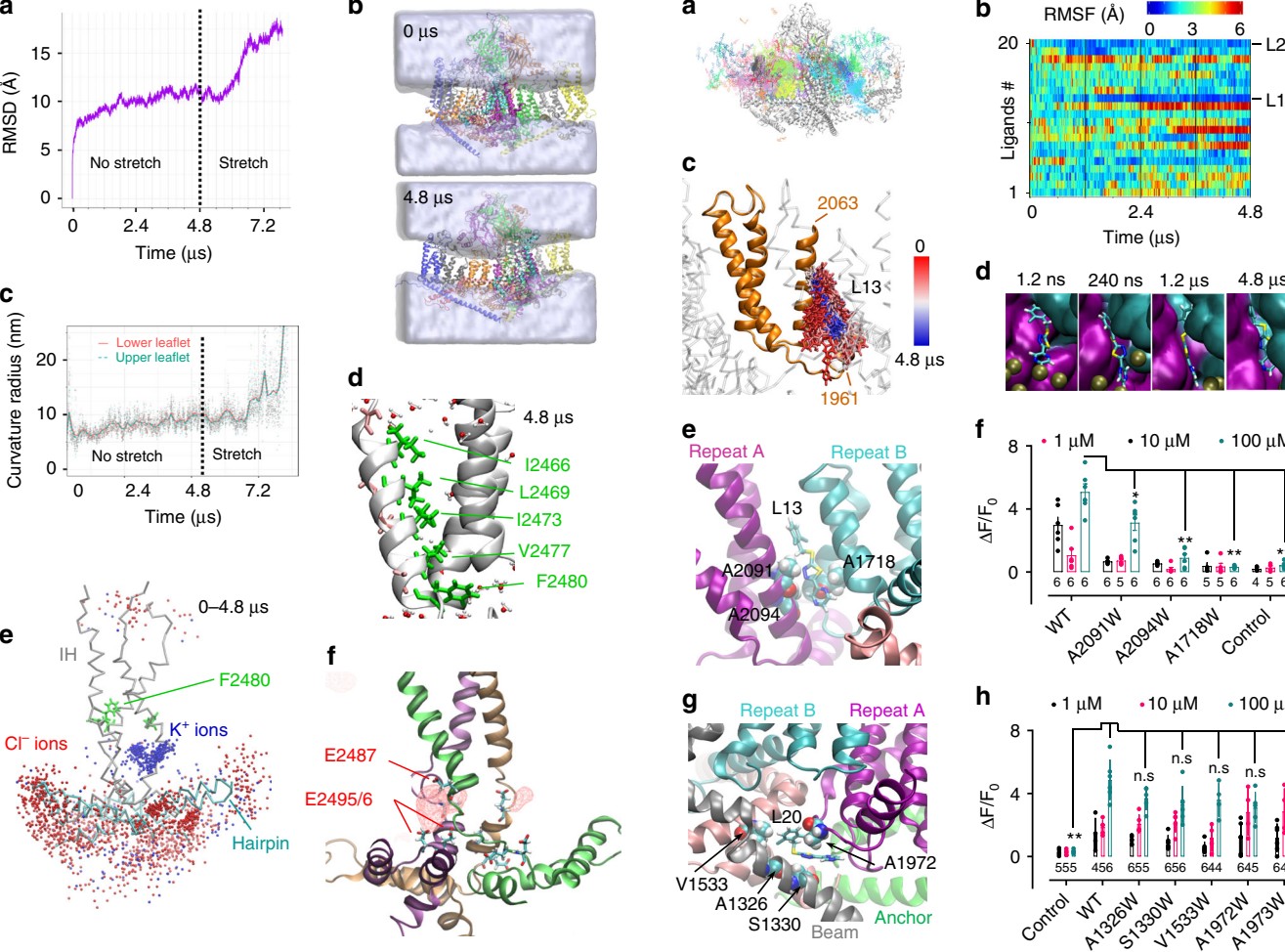

**Fig. 1** Membrane curvature and cation-selective pore fenestrations. **a** RMSD for protein backbone atoms during MD simulation. **b** Snapshots showing evolution of water density (shown as gray) around Piezo1 (shown as colored protein backbone). **c** Curvature radius plotted over time for the upper (red) and lower (cyan) leaflets. Dots represent individual time point curvature calculations, and lines represent LOESS fit with span of 4.7% of the total time using ggplot2 geom_stat function. **d** Detailed view of the channel pore showing the three inner helices and water molecules within 5 Å of protein backbone at $t = 4.8\,\mu s$. Side chains are shown in one inner helix for clarity. **e** Pore region showing accumulated K$^+$ (blue spheres) and Cl$^-$ (red spheres) ions within 5 Å of backbone and sampled every 24 ns during the first 4.8 μs (backbone shown at $t = 4.8\,\mu s$). IH, inner helices. **f** Accumulated K$^+$ ions density (red spheres)

**Fig. 2** Identification of a putative Yoda1 binding site and binding pathway. **a** Accumulated positions of all simulated ligands (every 24 ns) along 4.8 μs trajectory. The backbone is represented at $t = 4.8\,\mu s$. **b** RMSF time course for each simulated ligand (L1–L20) colored as indicated from the scale from lower (blue) to higher (red) values. **c** Accumulated L13 positions sampled every 24 ns from 0 (red) to 4.8 μs (blue) onto the backbone shown at 4.8 μs. Orange backbone corresponds to the 1961–2063 region. **d** Positions of L13 (licorice) sampled at the indicated time against the protein backbone (cyan: Piezo repeat B, magenta: Piezo repeat A, brown spheres: phosphate groups). **e** Positions of A1718, A2091 and A2094 and L13 inside the binding pocket sandwiched between Piezo repeats A (magenta) and B (cyan). **f** Dotted plot showing Yoda1-induced calcium signals in ΔPZ1 cells co-transfected with GC6 and the indicated constructs or transfected with GC6 only (control). **g** Position of L20 relative to surrounding small residues at 4.8 μs. **h** Dotted plot showing Yoda1-induced calcium signals in ΔPZ1 cells co-transfected with the indicated constructs and GC6 or transfected with GC6 only (control). In **f**, **h** the numbers indicate the number of independent experiments $n$ and each data point represents an average of at least 10 individual cells. Source data are provided as a Source Data file. Error bars = standard errors of mean values. Two-tails Mann–Whitney $U$-tests were performed to compare data distribution between WT and mutants/control for 100 μM Yoda1. Asterisks indicate standard p-value range: *, $0.01 < p < 0.05$; **, $0.001 < p < 0.01$; ***, $0.001 < p < 0.01$ and n.s. (non-significant): $p > 0.05$

to test if the loss of Yoda1-mediated calcium entry in the A2094W and A1718W mutants correlates with a loss of mechanosensitivity. The A2094W mutant produces pressure-dependent transient inward currents in a range of pressure similar to the wild-type (WT) channel (Fig. 3). In contrast, we were not able to detect pressure-activated currents from the A1718W mutant (Supplementary Fig. 2a). We next tested the effect of other A2094 and A1718 mutations on the channel's response to Yoda1 and Jedi2, another Piezo1 agonist, using calcium imaging (Fig. 4). In these experiments, we used the non-functional R1235A mutant as a negative control since this mutation abrogates mechanical opening by membrane stretch[39]. To our surprise, this mutant elicits robust calcium signals in the presence of both Jedi2 and Yoda1 (Supplementary Movie 2 and Fig. 4). These results indicate that at least one mechano-insensitive Piezo1 mutant retains normal chemical sensitivity to both agonists. In addition, among the

various mutations tested at positions 2094 and 1718, A1718G, A1718V, and A2094W retain partial Yoda1 sensitivity, while only A1718W and A1718G remain activatable by Jedi2. The lack of Jedi2 sensitivity in these mutants is surprising because Jedi2 is

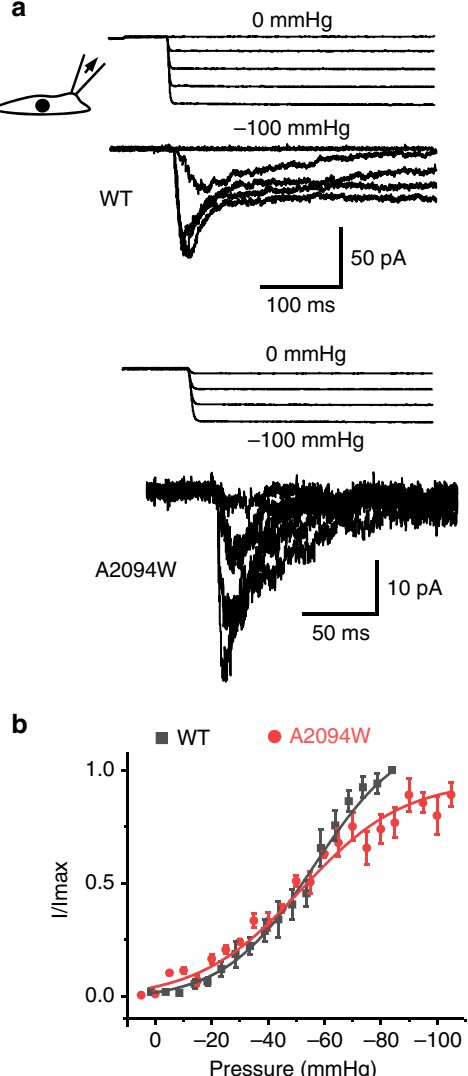

**Fig. 3** The Yoda1-insensitive A2094W mutant is mechanosensitive. **a** Example of pressure-elicited ionic current traces for mPZ1 WT and the A2094W mutant obtained in transfected ΔPZ1 cells clamped at −80 mV. For clarity, only few traces are shown. **b** Mean I/Imax values plotted for WT ($n = 8$) and the A2094W mutant ($n = 6$) for comparison. Data are collected from $n$ cells. Lines correspond to curve fitting using a Boltzmann equation. Source data are provided as a Source Data file. Error bars = standard errors of mean values

thought to interact with peripheral extracellular loops which are relatively distant from the putative Yoda1 binding site identified here[28]. In addition, these results indicate that the A1718W mutation does not completely abolish Piezo1 function since it retains some Jedi2-sensitivity.

We confirmed the absence of activity of the R2135A mutant in response to osmotic swelling (Supplementary Fig. 2b). We also tested whether the stretch- and Yoda1-insensitive mutant A1718W could mediate calcium intake in response to osmotic swelling. Although stretch-insensitive, the A1718W mutant elicits $Ca^{2+}$ signals similar to WT channel in response to osmotic shocks (Supplementary Fig. 2b). Together, these findings highlight the functional importance of this putative Yoda1 binding region for both the chemical and mechanical activation of Piezo1 and suggest the existence of several mechanical and chemical activation pathways that can be selectively eliminated by specific mutations.

**Yoda1 binding facilitates force-induced protein motions**. To further investigate whether the L13 ligand remains stably bound in absence of other ligands, we performed an additional 600 ns MD simulation wherein all ligands but L13 were removed (single-ligand simulation). This trajectory indicates that L13 indeed remains stably bound in absence of other Yoda1 ligands (Supplementary Fig. 3a). To study how Yoda1 binding promotes channel activation, we next performed a community network analysis derived from our single-ligand MD simulation trajectory. Communities are networks of residues that have stronger correlated motions between them than with other residues elsewhere in the protein. Thus, they generally correspond to protein parts or subdomains that move as relative rigid structural elements with respect to each other during conformational rearrangement[40–42]. Our community analysis indicates that in all subunits, regardless of occupancy by Yoda1, the L13 binding site is located between two independent communities, one formed by Piezo repeats BC (distal part of the arm) and one formed by Piezo repeat A (Supplementary Fig. 3b). We further noticed the loss of direct contact between the extracellular cap community and the repeat BC community in the Yoda1-bound subunit but not in the other subunits. Hence, these observations suggest that Yoda1 may increase the relative flexibility of the peripheral arm region relative to the central pore-cap region.

To test this hypothesis, we gradually increased membrane tension to a constant value of 40 mN m⁻¹ in our multi-ligand simulation. Although this value is above physiological stimulus intensity[22], we used it to speed up tension-induced conformational changes within microseconds. During stretching, the membrane curvature radius rapidly increased, allowing progressive flattening of the membrane (Fig. 1c). Concomitantly to membrane flattening, the arms rearranged, confirming their anticipated mechanosensory function (Supplementary Movies 3, 4)[25,26,29]. Despite on the large conformational changes observed in the Piezo1 arms, no major conformational changes are observed near the channel pore. Remarkably, the L13-bound arm (arm3) exhibits a larger RMSD (~20 Å) compared to other arms (~16 Å) (Fig. 5a, b). The direction of global conformational motion was extracted from random atomic motions by identifying collective displacement of protein alpha-carbons using a Principal Component Analysis (PCA). PCA analysis reveals that the main motion (75% of total protein motion) corresponds to an outward tilt of the distal part of the arms, a motion anticipated from the dome-shape structure of Piezo1 (Fig. 5c and Supplementary Movie 5)[26]. The maximum amount of tilt angle is larger in the L13-bound subunit (~80°) than in the other two subunits (~65° and ~68°), confirming the notion that Yoda1 binding increased mechanically-induced protein motions (Fig. 5d). The second principal component (9% of total protein motion) is a twist motion of the distal end of the arms, which shows different directions between the arms, consistent with the conformational heterogeneity observed in cryo-EM data (Fig. 5e)[27]. Projection of the protein alpha-carbon coordinates, at each timepoint, onto these two components reveals that the distal part of the arms undergoes a twist-tilt-twist motion sequence, illustrated by the U shape two-dimensional projection in Fig. 5f, suggesting that the large tilt of the Piezo1 arms induced by membrane stretch is enabled by a slight twisting motion.

## Discussion

To shed light on the mechanism by which the Piezo1 channel interacts with its Yoda1 agonist and responds to membrane stretch, a physiological stimulus for Piezo1, we performed a long all-atom unbiased MD simulation in the presence of the Yoda1 agonist and in absence and presence of increased membrane

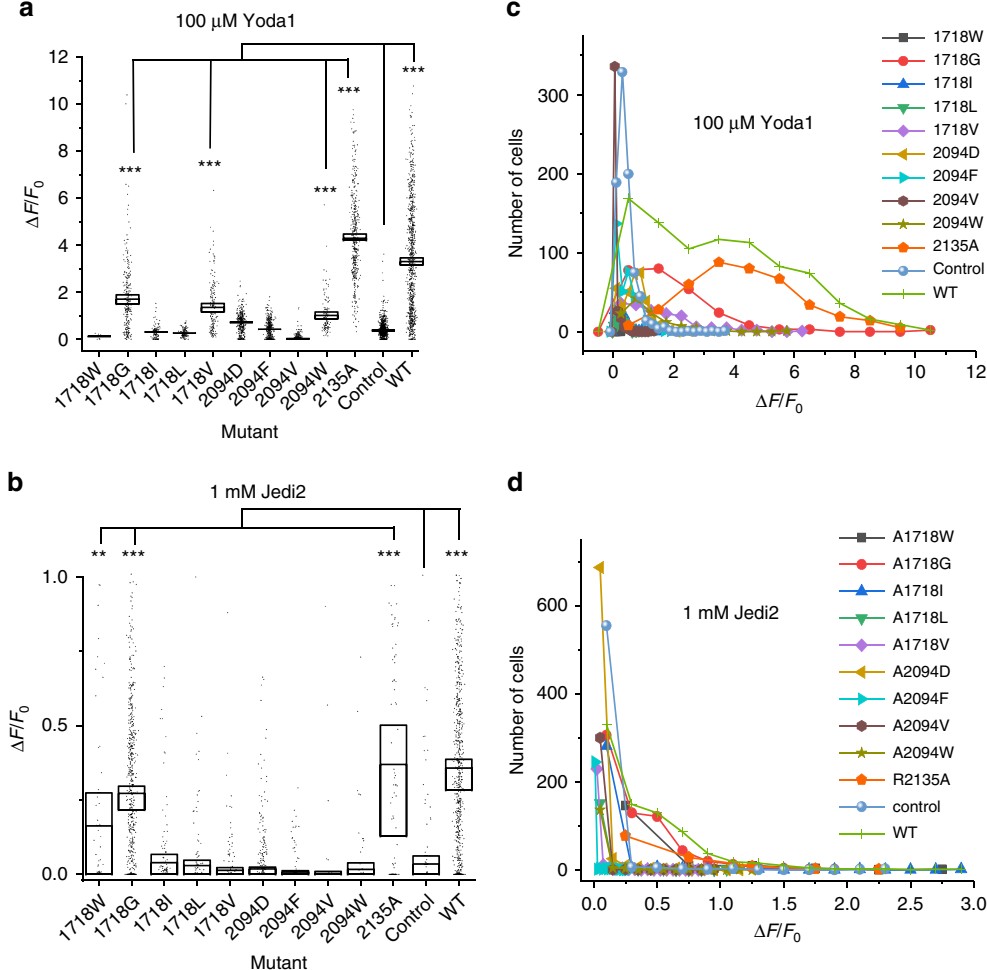

**Fig. 4** Coupling between chemical activation pathways in Piezo1. ΔPZ1 cells were co-transfected with GC6 and one of the indicated A1718 and A2094 mutant construct or with GC6 only (control). The relative amplitude of calcium signals ($\Delta F/F_0$) obtained by application of 100 μM Yoda1 (**a**) or 1 mM Jedi2 (**b**) is shown as a dotted box plot. For statistical analysis, cells from at least three independent experiments were pooled and considered as independent points (n values between 12 and 877). The box upper and lower limits represent standard error of mean values (shown as horizontal inner lines). Source data are provided as a Source Data file. Comparison of the mean values between WT/mutants and control was done using two-tails $t$-tests. Asterisks indicate standard $p$-value range. *, $0.01 < p < 0.05$; **, $0.001 < p < 0.01$; ***, $p < 0.001$ and n.s. (non-significant): $p > 0.05$. Statistical results are only shown for mutants exhibiting $\Delta F/F_0$ larger than control conditions. For clarity, the number of analyzed cells is from (a) and (b) are plotted as function of the range of $\Delta F/F_0$ values obtained with 100 μM Yoda1 (**c**) or 1 mM Jedi2 (**d**) for each tested mutant (gray rectangle: A1718W, red circles: A1718G, blue triangles: A1718I, green triangles: A1718L, purple diamonds: A1718V, gold triangles: A2094D, cyan triangles: A2094F, brown hexagons: A2094V, olive stars: A2094W, orange pentagons: R2135A, blue spheres: control and green crosses: WT)

tension. Our ~8 μs simulation recapitulates the anticipated inverted dome shape of the lipid bilayer with a curvature similar to that observed in proteoliposomes[26]. An interesting observation was the presence of potassium ions entering an intracellular pore vestibule through lateral fenestrations anticipated from cryo-EM studies[25–27]. We observed throughout our trajectory that negatively charged chloride ions were excluded from entering this vestibule, presumably due to the presence of glutamate residues essential for the Piezo1 cation selectivity near the intracellular side of the fenestrations[37]. Assuming that the entry of ions into these lateral fenestrations is not prevented by protein domains missing from the cryo-EM structures such as large intracellular loops, these results suggest that cations readily move in and out the pore vestibule even when the pore is non-permeant. This would suggest the main channel gate is located above the vestibule, at the level of a narrow constriction near F2480. This hypothesis is in good agreement with a recent study showing removal of the intracellular beam-to-latch linker region located below the vestibule does not constitutively open the pore,

indicating this intracellular region does not form a major permeation gate[43].

Our simulation identifies a Yoda1 binding site sandwiched between Piezo Repeats A and B. In principle, given the homotrimeric organization of Piezo1, three Yoda1 molecules can interact with Piezo1, i.e. one in each subunit. In our simulation, however, only a single binding event is observed. This is likely due to the fact that the 3-fold symmetry is not restrained during our simulation, allowing each subunit to sample a subset of conformations independently within the conformational ensemble. In addition, cryo-EM studies have reported the presence of asymmetricity between the channel subunits[25–27]. Additional binding events may hence theoretically be observable with longer simulations times. Mutations A1718W and A2094W in this binding site prevent or diminish the activatory effects of Yoda1. While the A2094W channel maintains its mechanosensitivity to stretch, we were not able to detect stretch-elicited currents from the A1718W mutant although this mutant responded to osmotic swelling, another mechanical stimulus for Piezo1[23,31]

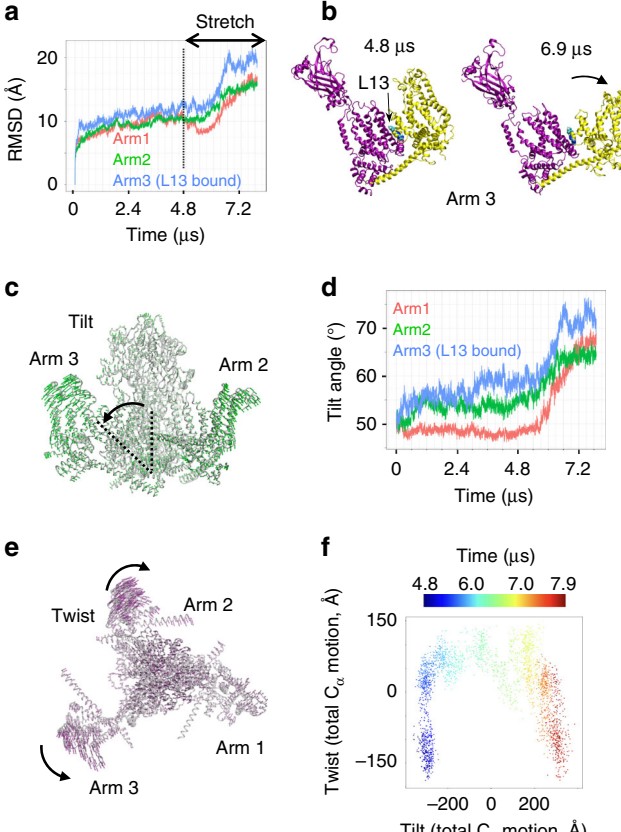

**Fig. 5** Yoda1 facilitates tension-induced arm motions. **a** RMSD for each Piezo1 arm along the full multi-ligand trajectory. **b** Trajectory snapshots of the Yoda1-bound subunit (arm 3) taken before (4.8 µs) or after (6.9 µs) stretch. **c** PCA representation of tilt motions. The green arrows indicate the direction of tilt motions along the stretch region of the trajectory. **d** Time evolution of tilt angle for each arm along the multi-ligand trajectory. **e** PCA representation of twist motions. The purple arrows indicate the direction of twist motions along the stretch region of the trajectory. **f** 2-dimensional projection of tilt and twist motions (represented as total $C_\alpha$ motion) colored as function of time during the stretch region of the trajectory. The 0 value in the tilt and twist axes represent the average position along the corresponding PCA motion

(Supplementary Fig. 2). If the substitutions of Ala residues to Trp in the binding site reduce or abolish Yoda1 binding, an independent validation of our binding site could be obtained by measuring Yoda1 binding affinity to purified Piezo1 mutants. However, this approach would not be effective if the loss of Yoda1 sensitivity was due to the binding of Yoda1 in a different, ineffective pose. This alternative scenario is possible since Dooku1, a Yoda1 analog, behaves as a competitive blocker of Yoda1, presumably interacting with the same binding site without activating Piezo1[44].

Two major mechanisms are typically used to interpret the effects of allosteric agonists such as Yoda1: the concerted and the sequential models. In the so-called concerted model, the affinity of the agonist for its binding site is higher for the open state than the non-conducting state: binding of the agonist hence shifts the open/close equilibrium towards the open state. In the alternative sequential model, binding of the agonist induces a local conformational change that allosterically propagates from the binding site to the active site (i.e., the channel pore). Although it is still unclear whether activation by Yoda1 follows one of these mechanisms, the fact that Yoda1 interacts with the purified

Piezo1 arm (mPZ1 residues 1–2190) with an affinity in the same order of magnitude as the $EC_{50}$ observed in cell-based assays (~10–50 µM)[28] suggests the affinity of Yoda1 for its binding site is relatively independent of the functional state of the channel. This observation is consistent with our simulation showing a spontaneous binding event to a non-conducting state. These observations favor a sequential model where the binding of Yoda1 induces conformational changes leading to pore opening.

Application of membrane stretch in our simulation reveals that Yoda1 binding enhances a twist-tilt-twist like opening motion of the arm. Based on the facts that (1) Yoda1 reduces the mechanical threshold for channel activation[30], (2) Yoda1 disrupts interactions between residue communities (Supplementary Fig. 1b) and (3) the arm motion is larger in presence of bound Yoda1 (Fig. 5a), we hypothesize that Yoda1 acts as a molecular wedge by decoupling the upper part of Repeats A from Repeat B, allowing the distal part of the arm to extend further in response to membrane tension (Fig. 6c). In contrast to the large protein motions observed in the arm regions, no significant gating motions were observed in the proximal pore region. Capturing these conformational changes on such large systems may require longer computational times which are still beyond the reach of even the most powerful supercomputers. Hence, the distal arm motions observed in our simulation may correspond to initial conformational steps that have not yet propagated to the pore. It is also possible that the full-length arms, acting as efficient mechanical transducers, are needed to open the pore. Our computational model only harbors Piezo Repeats A–C but six additional repeats (D–I) are predicted from cryo-EM studies and from the primary amino acid sequence[26]. MD simulations adding repeats D–F may reveal important clues regarding the mechanism of mechanotransduction in Piezo1. Future cryo-EM structures solving the most peripheral Repeats (G–I) as well as other large soluble loops unresolved in the current structures will be needed to create a native, full-length computational model of Piezo1. In addition, Piezo1 spontaneously opens in asymmetric, but not symmetrical, artificial bilayers and specific membrane lipids have been found to either inhibit or promote Piezo1 activity[23,30,45,46]. Hence, manipulation of the chemical composition of the membrane in computer simulations, such as creating a native-like asymmetric bilayer, may facilitate in silico spontaneous gating transitions towards the open state upon stretch. Our current simulation of a reduced Piezo1 model in a symmetric POPC membrane represent a starting point for future computational studies.

Electrophysiology evidence showed Yoda1 acts as a gating modifier by reducing the channel's mechanical activation threshold by stretch and poking[30]. This suggests activation by Yoda1 and by mechanical stimuli are coupled, as reflected in our wedge model. However, many of the mutations characterized in this study and others abolish Piezo1 activation by chemical stimuli but not mechanical stimuli or vice versa. For instance, the R2135A mutation, located in the anchor region between the pore and repeat A, disrupts mechanical activation by stretch[39] and swelling (Supplementary Fig 2b) but does not affect sensitivity to Yoda1 and Jedi2. The beam mutations L1342A/L1345A prevent chemical activation by both Jedi1 and Yoda1 and mechanical activation by cell-poking but maintain stretch sensitivity[28]. Likewise, the A2094W mutants exhibit a loss of sensitivity to both Jedi2 and Yoda1 (Fig. 4) but not to membrane stretch (Fig. 3). These mutant phenotypes would suggest that, beyond our simplistic wedge model, chemical and mechanical stimuli may be sensed by independent transduction pathways. Future studies will be needed to elucidate how these mutations can selectively affect chemical or mechanical activation.

Although both Jedi2 and Yoda1 interact with the Piezo1 arm, Jedi2 is negatively charged at physiological pH and hence is

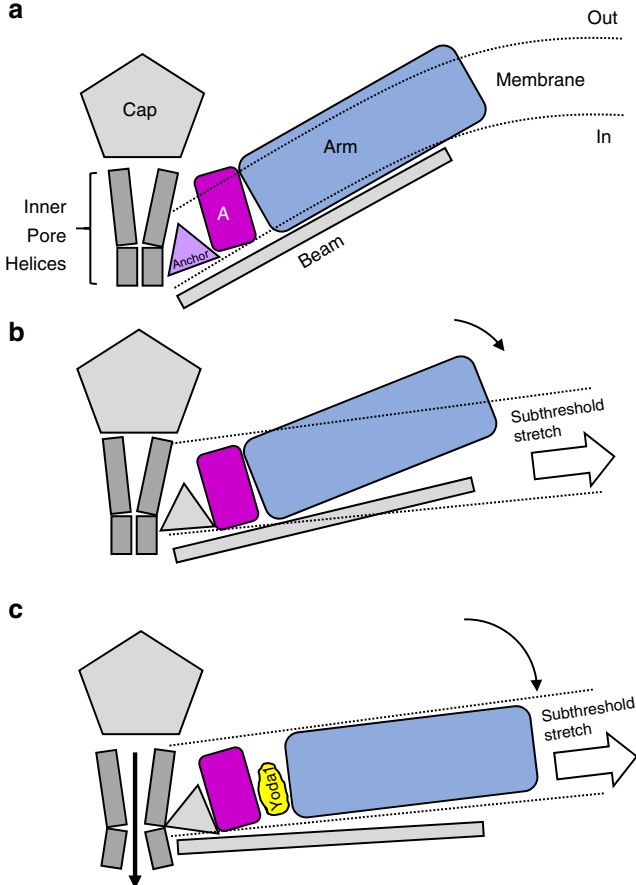

**Fig. 6** A molecular wedge mechanism for Yoda1-mediated Piezo1 activation. **a** In absence of membrane tension, the Piezo1 arm (blue) is in a "flexed" position (only one arm is represented for clarity). **b** In presence of a sub-threshold stimulus, the arm is slightly extended due to the flattening of the lipid bilayer but not enough to open the pore. **c** By binding between Repeat A (magenta) and the N-terminal part of the arm, Yoda1 acts like a wedge by decoupling these two domains, hence increasing tension-induced arm extension. This wedge-like effect ultimately leads to channel opening in the presence of sub-threshold stimuli. The mechanism by which the lever motion opens the channel gate is unknown

thought to interact with distal extracellular loops[28]. In contrast, according to our simulation data, the hydrophobic Yoda1 diffuses quickly (within 2 nanoseconds) from the solvent to the membrane and enters its binding site from the intracellular side of the membrane, as anticipated from excised-patch electrophysiology experiments[28]. The fact that the sensitivity of Piezo1 to both Yoda1 and Jedi1/2 can be simultaneously abolished by the same point mutation is puzzling. It is possible that, although acting from distinct protein regions, both agonists somehow modulate a common long-distance allosteric transduction pathway traversing a large fraction of the Piezo arm as suggested previously[28].

As shown in this study and others, the agonist effects of Yoda1 are highly sensitive to protein mutations as well as small modifications of the Yoda1 chemical structure[30,44]. It is puzzling to reconcile the stringent nature of the Yoda1-Piezo1 interaction with a relatively simple wedge-like mechanism. It is possible that the modulatory effect of Yoda1 necessitates a specific binding pose placing specific chemical groups in an optimal orientation. Structure-activity relationships suggest that the (2.6-dichlorobenzyl)thioether group of Yoda1 is critical for binding while the pyrazine and thiadiazol groups are important for its agonist effect[44]. The Yoda1 binding site revealed in this study will enable

future simulations and structure-functions studies that will help understand the structure-activity relationships of Yoda1 and its analogs. Such studies will also be invaluable to elucidate the origin of the Piezo1 vs. Piezo2 selectivity of Yoda1.

In conclusion, our study identifies a potential allosteric Yoda1 binding site in Piezo1 and suggests a mechanism wherein Yoda1 binding facilitates force-induced protein motions through a wedge-like mechanism. This work will help the rationale design of highly-sought-after isoform-selective Piezo modulators and pave the way towards understanding how Piezo channels are modulated by chemical and mechanical stimuli.

## Methods

**System preparation for MD Simulations**. Two recent cryo-EM structures of mouse Piezo1 (PDBID: 6BPZ and 6B3R) obtained at ~3.8 Å resolution were first aligned and compared the amino acid sidechain assignment. Since the two structures are consistent, the latter one (6B3R) was used to build the all-atom model. We added the missing loops that are shorter than 20 amino acids using the loop remodeling function of Molecular Operating Environment[47]. The protein was then embedded in a phosphatidylcholine (POPC) membrane and solvated in TIP3P water molecules and 150 mM KCl (432 $Cl^-$ ions and 399 $Na^+$ ions to ensure electroneutrality of the whole system) using CHARMM-GUI website[48]. The full atomic model built from the cryo-EM structure with bilayer and solvent contains 1 million atoms, which exceeds the size limit on ANTON2 supercomputer (700,000 atoms). Because this system was intended to capture spontaneous Yoda1 binding, we removed the most peripheral repeats D–F, which were not anticipated to contain the binding site[31]. In the final reduced model, each subunit of the Piezo trimer contains four segments (Supplementary Fig. 1) including residue 1131–1365 (Piezo repeat C and beam), 1493–1578 (clasp), 1655–1807 (repeat B), and 1952–2546 (repeat A, anchor, TM37, cap, TM38, and CTD). A coiled peptide under CTD domain, residue 1403–1417 (latch), although solved in one of the cryo-EM structures (PBD 6BPZ), is not included. This is because (1) there are substantial residues missing on both end of this short peptide which would require additional constraints during the MD simulation and (2) a recent study indicates the latch region does not form a major permeation gate[43]. The PROPKA[49] was used to determine protonation states of titratable residues based on an environment at pH 7. In order to capture the spontaneous binding of Yoda1, we inserted 20 Yoda1 molecules (labeled L1–L20) into our system to ensure sufficient sampling during ANTON2 simulation. This corresponds to a ligand concentration of about 5 mM, which is lower than the concentrations used in spontaneous binding simulations[32,34,50]. CHARMM36 parameter sets were used throughout, including the protein, ions[51,52], POPC lipids[53], and TIP3P for water[54]. CHARMM general force field (CGenFF)[55,56] was used for Yoda1 molecule. The terminal amino acids of each segment are capped using acetylated N-terminus (ACE) and methylamidated C-terminus (CT3) blocking groups. We obtained a model with 602,203 atoms (recommended size for maximum efficiency on the ANTON2 machine) with a system box size of $190.1 \times 190.1 \times 177.5$ Å$^3$.

**Equilibration**. The resulting starting structure was first run through the equilibration using pemed.CUDA module of AMBER16[57]. This consisted of an initial minimization followed by a six-stage thermal equilibration phase. Following the final thermal equilibration stage, 10 ns of 'unrestrained' equilibration was conducted. Here, unrestrained refers to the release of membrane and protein backbone-conformation restraints. A 30 Å flat-bottom harmonic restraint with spring constant of 0.12 kcal mol$^{-1}$ Å$^{-1}$ was added between center of mass of the residue 1273–1299 of the beam and the center of mass of the residue 2488–2543 at the bottom of the pore throughout all the simulations. This restraint is to prevent the C-terminal of the beam drifting too far due to the absence of an unstructured loop sequence from the cryo-EM data. During the first 20 ns, all 20 Yoda1 molecules quickly diffuse from water into the lipid bilayer. Since one primary goal of this simulation procedure was to identify potential binding sites for Yoda1, to ensure the sufficient sampling of Yoda1 ligand near the protein region, a 45 Å radius flat-bottom harmonic spherical restraints (centered at the center of mass of residues 1952– 2190 with spring constant of 0.12 kcal mol$^{-1}$ Å$^{-1}$) were applied to limit the diffusion of six ligands around each Piezo monomer throughout all the simulations. Since the volume of these restraints was chosen to be sufficiently large such that the entire transmembrane region of the Piezo is included, these restraints are there only to ensure that Yoda1 molecules would not drift into distant regions of the surrounding membrane environment without any bias on the interaction between Piezo1 and Yoda1. Due to system size, a total of 7500 cycles of minimization were run, with 5000 steepest decent cycles followed by 2500 conjugate gradient cycles. For this stage and all following, a non-bonded cut-off of 12.0 Å plus a 10.0 Å force switching range was employed. Both positional and internal conformational restraints were applied. Positional restraints were applied to all atoms of all protein residues with a 10.0 kcal/mol/Å harmonic spring constant while phosphate atoms of membrane POPC residues were restrained with a force constant of 2.5 kcal mol$^{-1}$ Å$^{-1}$. Dihedral restraints were also employed to ensure retention of backbone phi-psi angles

during minimization, with 250 kcal mol$^{-1}$ radian$^{-2}$ force constant. Following minimization, a 25 ps NVT solvent heating simulation was run to attain a target temperature of 310.15 K. For this stage through the fifth equilibration stage, a time step of 1 fs was employed. The same positional and internal restraint setup was applied as was utilized during the previous minimization stage. For all simulation stages, temperature control was accomplished using a Langevin thermostat with a gamma parameter (friction coefficient) of 1.0 ps$^{-1}$. The SHAKE algorithm was used to constrain bonds involving hydrogen. The second equilibration stage followed an identical setup with the sole exception that the force constant of the protein positional restraints was reduced to 5.0 kcal mol$^{-1}$ Å$^{-1}$ (down from 10.0 in the previous stage). The force constant of the dihedral phi-psi restraint was also reduced to 100 kcal mol$^{-1}$ radian$^{-2}$ (down from 250). The first and second stage equilibrations were each run for a duration of 25 ps each. Starting with the third equilibration stage, the duration of simulations was raised to 100 ps and an NPT ensemble was employed, with pressure regulation accomplished by utilization of a semi-isotropic Monte-Carlo barostat with a target pressure of 1.0 bar and constant zero membrane surface tension. For the third stage, force constants for protein and membrane positional restraints were reduced to 2.5 and 1.0 kcal mol$^{-1}$ Å$^{-1}$, respectively (down from 5.0 and 2.5). The dihedral phi-psi protein backbone restraints were also dropped to 50.0 kcal mol$^{-1}$ radian$^{-2}$ (down from 100.0). In the fourth stage, positional restraint force constants for protein and membrane were reduced to 1.0 and 0.5 kcal mol$^{-1}$ Å$^{-1}$, respectively (down from 2.5 and 1.0). In the fifth stage, positional restraint force constants for protein and membrane were reduced to 0.5 and 0.1 kcal mol$^{-1}$ Å$^{-1}$ (down from 1.0 and 0.5) and phi-psi backbone dihedral restraint force constants were reduced to 25.0 kcal mol$^{-1}$ radian$^{-2}$ (down from 50.0). For the sixth and last equilibration simulation, membrane positional restraints and phi-psi backbone dihedral restraints were removed entirely while protein positional restraints were reduced to 0.1 kcal mol$^{-1}$ Å$^{-1}$ (down from 0.5). Simulation timestep was also raised to 2.0 fs as it would remain for all future simulations. Finally, a fully unrestrained equilibration production run was conducted for a total of 10 ns (all other parameters were identical to the sixth stage of equilibration).

**Microsecond molecular dynamics simulation.** For zero tension simulation, the endpoint from equilibration was uploaded to the ANTON2 supercomputing cluster and converted from AMBER topology and coordinate format into an equivalent DESMOND chemical model format using the viparr modeling and simulation suite. An ark file was constructed to match as closely as possible the simulation parameters employed under the AMBER equilibration stage. Due to the size of our system, nearly seventy thousand atoms, it was necessary to raise the energy and force scale limits from the default of 1000–5000 kcal mol$^{-1}$ or kcal mol$^{-1}$ Å$^{-1}$, this apparently is a relatively common requirement. Similarly, due to the size of this system and the relatively long timescale being simulated, a relatively low trajectory output frequency of 1.2 ns was chosen.

As with the previous AMBER equilibration step, the ANTON2 simulation was run under an NPT ensemble. Pressure regulation was accomplished via the MTK barostat, set to maintain 1 bar of pressure, with a tau (piston time constant) parameter of .0416667 ps and reference temperature of 310.15 K. The barostat period was set to the default value of 480 ps per timestep. Temperature control was accomplished via the Nosé-Hoover thermostat with a tau (time constant) parameter of .0416667 ps. For the purpose of the barostat reference, the mts parameter (substepping) was set to 4 timesteps, while for temperature control purposes a value of 1 timestep was used. The thermostat interval was set to the default value of 24 ps per timestep. The timestep for the simulation was chosen to be 2 fs. This is slightly slower than the standard 2.5 fs per timestep but was found to be necessary to ensure simulation stability in prior exploratory testing for this system. Due to the presence of a lipid membrane, semi-isotropic pressure scaling was required. For this zero-tension simulation, the tension reference was set to 0.

To prevent incidental drift, center of mass motion removal was turned on. Respa was employed for all ANTON2 simulations with the default bonded interval of 1 timestep, far non-bonded interval was set to 2 timesteps and non-bonded near interval set to 1 timestep. The slightly more frequent far non-bonded interval was again chosen to ensure simulation stability as the lower default frequency seemed to induce more frequent simulation restarts thus making the higher frequency yield better overall throughput due to the significant overhead in restarting simulations.

Simulations of membrane stretching performed on ANTON2 followed the same protocol as zero tension simulations with the exception that membrane tension tensor was set to a fixed value. Stretching was conducted over three stages, starting from the end of the 4.8-microsecond zero tension simulation. In the first stage, tension was increased from 0 to 20 mN m$^{-1}$ over a series of twenty 60 ns simulations. At each step in the series, tension was set to 1 mN m$^{-1}$ higher than the previous step. The second stage, which started from the endpoint of the first stretching stage, consisted of a series of ten steps wherein the tension was increased by 2 mN m$^{-1}$ in each consecutive 30 ns step (up to a total of 40 mN m$^{-1}$). Finally, the last stage consisted of 600 ns of additional simulation time at the final tension of 40 mN m$^{-1}$ starting from the endpoint of the second stage. All the input files and trajectories are free to download from ANTON2 repository.

**Determination of membrane curvature.** Trajectory data from the microsecond MD simulations was first preprocessed using the *cpptraj* program from the AMBER suite. All solvent and ion molecules were first stripped from the system in order to reduce storage and computational power requirements for processing. The *autoimage* command was then run using the largest segment of the first subunit of the Piezo trimer as the anchor molecule. This ensures that protein structure remains as one contiguous unit in all frames of the trajectory. This is required to accommodate the default behavior of wrapping all atoms to the central unit cell in the output trajectory data due to the periodic boundary condition, which would complicate further analysis. The resulting trajectory data containing membrane and protein components was then processed using the *pytraj* python API from the AMBER suite. This allowed direct access to *cpptraj* subroutines from within a python environment as provided by Jupyter Notebook. The center of mass for each lipid headgroup was then computed for each lipid at each frame of the trajectory and stored into a pair of arrays corresponding to the upper and lower membrane leaflets respectively. The resulting headgroup center of mass data were then analyzed to determine parameters for interpolation of the heights of membrane headgroups to a two-dimensional grid. Grid spacing was chosen to be 4.6 Å which corresponds to about twice the molecular radius of a phosphate ion as a rough proxy for the radius of lipid headgroups. The grid dimensions were then chosen such that all headgroup center of mass coordinates, as measured over all frames of trajectory, would fit within the grid with a padding of one grid spacing on each edge. This yielded a grid with 52 gridpoints along the *x*-axis and 50 grid points along the *y*-axis. This grid extending from roughly −118.8 Å to +114.8 Å in the x dimension and from −114.7 Å to +108.6 Å in the y dimension. Prior to interpolation to the grid described above, the eight adjacent *x*–*y* periodic images of the computed lipid center of mass data were computed for each frame and appended to the computed center of mass data for each given frame. This combined periodic data were then interpolated onto the two-dimensional grid described above using bi-linear interpolation. This interpolation was computed using the *sp.interpolate.griddata* function from the SciPy Python library. Next, the linear interpolated data was further smoothed using a two dimensional 'local mean' smoothing kernel with a size of 13 grid units, as implemented under the *sp.ndimage.filters.uniform_filter* function from the SciPy library. The above kernel size corresponds to the one quarter of the geometric mean of the grid dimensions, rounded up. The 'wrap' mode was employed to resolve edge conditions. This yielded a two-dimensional grid representation of membrane surface height at each frame of the trajectory for both the upper and lower leaflet. Next, a bivariate radial Gaussian was fitted to the resulting grid data at each frame using the *sp.optimize.curve_fit* function. The form of this fitted function was given as:

$$z_0 + z_h * e^{-\left(\frac{\sqrt{(x-r_x)^2+(y-r_y)^2}}{\sigma}\right)^2} \tag{1}$$

Here, the z offset of the Gaussian ($z_0$) effectively models the plateau height of the membrane, $z_h$, the z scaling factor, models the depth of the indentation induced by piezo, $r_x$ and $r_y$, model the location of the center of the induced indentation. σ is standard deviation of the Gaussian.

Finally, the radius of curvature at the peak of these fitted Gaussian distributions was computed using the formula:

$$d = \frac{-2z_h}{\sigma^2} \tag{2}$$

$$R = \frac{(1+d^2)^{\frac{3}{2}}}{d} \tag{3}$$

Where, R is the radius of curvature at the center of the fitted Gaussian and d is the evaluation of the first and second derivative of the fitted gaussian function at $x = r_x$ and $y = r_y$. This radius of curvature data was plotted using the ggplot2 package in R to allow visual inspection of changes in membrane curvature throughout the trajectory.

**Community analysis.** The community analysis was also done based on the Pearson correlation coefficients of alpha carbon motions from the 600 ns single Yoda1 trajectory and the Girvan-Newman algorithm implemented in the NetworkView program in VMD[58–60].

**Focused PCA of Piezo1 arm motion under membrane tension.** Principal component analysis was performed using the backbone alpha carbons of all simulated piezo residues for the portion of simulation corresponding to the second stage of membrane stretching. Calculation of principal component modes was carried out using the pytraj module of the AMBER modeling and simulation package. The resulting principal component data was exported to Norma Mode Wizard format for subsequent visualization using the VMD visualization program.

**Molecular cloning.** Point mutations were introduced into the mouse Piezo1 cDNA cloned into a pCDNA3 plasmid (a gift from Dr. Ardem Patapoutian, Scripps Research) using Gibson Assembly or Q5 Site-Directed Mutagenesis (New England Biolabs). Primers used in this study are listed in Supplementary Note 1. All constructs were verified by automated Sanger sequencing (Genewiz).

**Cell culture and transfection**. HEK293T-ΔPZ1 cells (a gift from Dr. Patapoutian, Scripps Research) were cultured in standard conditions (37 °C, 5% $CO_2$) in a DMEM medium supplemented with Penicillin (100 U mL$^{-1}$), streptomycin (0.1 mg mL$^{-1}$), 10% sterile fetal bovine serum and 1X MEM non-essential amino-acid without L-glutamine. All cell culture products were purchased from Sigma-Aldrich. Transfection was done on cells with a passage number lower than 25 using FuGene6 (Promega) using the manufacturer's instructions. For fluorescence experiments, cells seeded on clear-bottom black 96-well plates (Costar) were transfected at ~50% confluence 2–4 days before the experiment. For electro-physiology experiments, cells seeded on Matrigel-coated coverslips were transfected at ~10% confluence 12–36 h prior recordings.

**Calcium imaging**. To measure Piezo1-mediated calcium entry, cells were co-transfected with a pCDNA3.1 plasmid encoding the fluorescent calcium indicator GCaMP6m (GC6). Transfected cells were washed twice with Hank's Balanced Salt Solution (HBSS) and incubated at least 10 min in an incubator (37 °C, 5% $CO_2$). Cells were then placed on an inverted fluorescence microscope (Nikon Eclipse) and illuminated by a 100 W mercury lamp through a ×20 objective. Fluorescence images were obtained using a standard GFP filter set and acquired at 1 frame s$^{-1}$ using a Nikon Digital Sight camera and the Nikon Digital Element D software. Yoda1 (Sigma-Aldrich) was dissolved at 10 mM in dimethylsulfoxide (DMSO) and various Yoda1 solutions were made by diluting the Yoda1 stock solution into HBSS. During recordings, 100 µL of a 2× Yoda1 solution was added to the cells at $t = 10$ s. Total DMSO concentration was kept equal to or below 1% for all tested Yoda concentrations. A similar protocol was used for Jedi2 (purchased from MolPort) where the final agonist concentration was 1 mM and the stock con-centration was 100 mM (in DMSO). For osmotic shock experiments, a hypotonic saline solution (where 150 mM NaCl was replaced by 5 mM NaCl) was added to the cells as described elsewhere[31]. Regions of interest corresponding to individual or clustered cells were selected either manually or using an in-house MATLAB script (Supplementary Note 2). The maximal values of relative change in fluores-cence intensity $F$ ($F_{max} - F_{t=0}/F_{t=0}$, or $\Delta F/F_0$) were calculated for each selected region following application of chemical stimuli. Except for Fig. 4a, b, the $\Delta F/F_0$ values were averaged across the field of view for each movie and each movie was treated as an independent measurement. For Fig. 4a, b, individual cells were counted as independent measurements and pooled across at least 6 movies.

**Cell-attached patch-clamp recordings**. During recordings, the membrane potential was zeroed using a depolarizing bath solution containing 140 mM KCl, 1 mM $MgCl_2$, 10 mM glucose and 10 mM HEPES pH 7.3 (with KOH). Fire-polished patch pipettes with a diameter of 2–3 µm and resistance of 1–5 MΩ were filled with a recording solution containing 130 mM NaCl, 5 mM KCl, 1 mM $CaCl_2$, 1 mM $MgCl_2$, 10 mM TEA-Cl and 10 mM HEPES pH 7.3 (with NaOH). Stretch-activated currents were recorded in the cell-attached configuration after seal for-mation using an Axopatch 200B amplifiers (Axon) and a high-speed pressure clamp (HSPC-1, ALA Scientific Instruments). The membrane potential inside the patch was held at −80 mV. Data were recorded at a sampling frequency of 10 kHz and filtered offline using a 1–2 kHz lowpass digital filter and a digital filter to remove electrical interferences (pClamp, Axon). Peak current amplitudes were determined by adjusting the baseline prior the pressure pulse. The relative peak current values and standard error are from averaging recordings from n transfected cells. Fitting the plots of I/Imax vs. pressure was done using a Boltzmann function (OriginPro 2019).

**Statistics**. Measurements were taken from independent samples and no mea-surement consisted of repeats from previously measured samples. To test statistical difference between mean values, we performed two-tails Mann–Whitney $U$-tests between WT mPZ1 and each mutant/control for 100 µM Yoda1. For Fig. 4, each analyzed cell was treated as independent region of interest and pooled between several experiments and statistical analyzes were done using two-tails $T$-tests assuming unequal variance among tested series. The exact $n$ values are reported in the provided Source Data file. All errors bars = s.e.m. Asterisks indicate standard $p$-value range. *, $0.01 < p < 0.05$; **, $0.001 < p < 0.01$; ***, $p < 0.001$ and n.s. (non-significant): $p > 0.05$.

**Reporting summary**. Further information on research design is available in the Nature Research Reporting Summary linked to this article.

## Data availability

Data supporting the findings of this manuscript are available from the corresponding authors upon reasonable request. A reporting summary for this Article is available as a Supplementary Information file. The source data underlying Figs. 2f–h, 3b, 4a, b and Supplementary Fig. 2b are provided as a Source Data file.

## Code availability

The MATLAB code used for our automated fluorescence imaging analysis is available as Supplementary Note 2. The pdb file of the all-atom mouse Piezo1 model and simulation input files used for this study are available at: https://github.com/LynaLuo-Lab/Piezo1-all-atom-model.

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

## Acknowledgements

This work was supported by Pittsburgh Supercomputing Center ANTON2 allocations PSCA17006P and PSCA18007P (to Y.L and J.J.L.), NSF XSEDE research allocation MCB160119 (to Y.L. and J.J.L), R21 NINDS-NIH grant 1R21NS101384–01A1 (to J.J.L.), start-up funds from Western University of Health Sciences (to J.J.L. and Y.L.) and an intramural research grant from Western University of Health Sciences (to J.J.L. and Y.L.).

## Author contributions

Y.L., W.M.B-S., H.Z., W.J., and C.P. designed and performed computer simulations; J.J.L. and A.D.O. designed and performed experiments; A.D.O. created the MATLAB code; all authors analyzed data; Y.L. and J.J.L designed the project and wrote the paper with inputs from all authors.

## Competing interests

The authors declare no competing interests.
