## [Peer Review File · Nature Communications]

Reviewers' Comments:

Reviewer #1:

Remarks to the Author:

The paper is about the determination of a possible mechanism of activation of mechanosensitive ion channel Piezo1 by agonist Yoda1. The main investigation method is based on microsecond-long all atoms molecular dynamics (MD) simulations, with additional evidences coming from calcium imaging spectroscopy and electrophysiological techniques. MD simulations of the system (Piezo1 + membrane + solvent) were carried out in different conditions (with and without tension, with and without ligands etc.)'.

The authors conclude that Yoda1 binds to a specific hydrophobic pocket sandwiched between Piezo1 Repeats A and B" and that "binding may alter a relative conformational motion between the pore region (that includes the most proximal repeat A) and the remaining N terminal part of the arm."

This suggested outcome is intriguing, and the authors support their conclusion performing a community analysis on a simplified system - with all ligands, except L13, removed - putting in evidence that L13 acts as a hinge between the two repeats.

The paper presents a significant collection of arguments sustaining the authors conclusions.

From the technical point of view the methods employed are presented flawlessly, with even an excess of specific details for the MD part. This could be certainly helpful in case other workers would like to attempt reproducing the observed results (although the availability of the computational tool employed, ANTON, remains limited). The statistical analysis of the experimental evidences presented in the paper is, as far as I can judge, correct.

The Discussion session is sketchy and could probably be improved with additional comments. For instance, is it possible to estimate quantitatively the nature of the motions induced by tension in the presence of Yoda1. The qualitative assertion "a large tilt, or extension of the distal part of the arm" could be likely be made more quantitative (amplitude of motion and if possible estimate of time-scale).

In any case the paper presents a persuasive and convincing picture which is certainly of interest to a significant audience, especially biophysicists and computational biochemists. I recommend publication in Nature Communications.

Reviewer #2:

Remarks to the Author:

Using molecular dynamic simulations, the authors proposed a mechanism that the small molecule Yoda1 activates mechanosensitive Piezo channel by binding to an allosteric pocket like a wedge. Although the major finding is from MD simulations, the authors did a nice job validating the proposed binding pocket, as well as excluding another possible binding site, by mutagenesis, calcium imaging, and electrophysiology. In addition, the simulations showed several features that agree well with literature: a. potassium ions accumulated near the negatively-charged fenestrations, b. the lipid bilayer forms a dome after equilibration, and c. the channel flattens with the lipid bilayer when tension is applied. This study is novel. How Yoda1 works on Piezo1, either through direct binding or via modifying the local lipid environment, is under debate in the field and would be important to help understand the mechanism of Piezo gating in general. Though interesting, there are several points that need to be addressed before publication in Nature Communications.

1. In the simulation system, 20 Yoda1 ligands were added. Based on Figure 2a and Movie S1, the 20 ligands appeared sampling the entire trimeric Piezo channel. Then why is there only one ligand

binding to the L13 site in one of the three subunits (arm 3), i.e. why do other ligands not bind to the equivalent L13 sites in other subunits? Lacroix et al. (ref. 30) suggested that only one Yoda1-binding subunit is sufficient for Yoda1-mediated activation, but it is not clear whether the activation is through a concerted or sequential mechanism. So it would be interesting to see the simulations with 3 Yoda1 ligands bound to the L13 sites of all three subunits. Maybe only then would the pore open?

2. The two Piezo paralogs have high sequence identity, yet Piezo2 is insensitive to Yoda1. Could it be explained by the mechanism presented here? Although the structure of Piezo2 is not available, sequence alignment and topology prediction could be well done with reference to Piezo1. Many regions are highly conserved between Piezo1 and Piezo2, including the helices containing A1718, A2091, and A2094 (corresponding residues A, G, and A in Piezo2). So it appears that the Yoda1 binding pocket might also be present in Piezo2. Then what would be the mechanism of the insensitivity of Piezo2 to Yoda1?

3. Due to the technical limitation of the supercomputer, which is understandable, the three peripheral repeats D-F were removed from the simulations. However, including these three repeats, which contribute significantly to the overall curvature of the channel, would likely drive the simulation more to an open state. The authors did mention that full-length arms might be required to open the pore in the simulations. But not only the structurally unresolved ones, but these three structurally available repeats could also well be essential. It is important to state that.

4. The authors chose POPC as the lipid component of the membrane without giving any reason. This is a commonly used lipid, but Piezo and Yoda1 may actually require an optimized set of lipids to work properly. Bending moduli, intrinsic curvature and symmetry of the lipid bilayer could all be critical to open this unusually curved channel. The authors may take advantages of simulations to explore different lipid compositions in Piezo gating. The actual work may be well beyond the scope of this manuscript, but it is worth some solid discussion here.

5. Related to the effect of lipid on Piezo, during the simulations, the authors carefully set the restraints so that Yoda1 molecules are not far in the surrounding membrane environment. What if Yoda1 could also work by modulating membrane property since it readily diffuses into the lipid? Could this be captured by the simulations? Or have the authors already tested that and excluded this possibility? It would be important to know if this is the case.

6. The authors presented PCA analysis of the subset corresponding to the ten 30 ns trajectories over which membrane tension increased from 20 to 40 mN/m (time points from 6.3 to 6.9 μ s in Figure 4f, right?) because this subset yielded the largest global changes in structure. Since other subsets were also subjected to the same analysis, did they give consistent, just smaller, changes? Or did they show different patterns? It would be interesting to know if Piezo moves differently under different tension regimes. Especially if the authors would like to claim the twist-tilt-twist motion sequence, the two-dimensional projection analysis should include earlier time points since the tilt angle started to increase between 5.4 to 6.0 μ s according to Figure 4d.

Minor points:

1. This manuscript only discussed Yoda1, not other Piezo1 chemical activators such as Jedi1 and Jedi2, which work through the extracellular side, thus likely a different mechanism compared to Yoda1. Therefore, the title might be too broad which implies a general mechanism for all Piezo1 chemical activators.

2. Line 80, the mutant A1718A should be A1718W.

3. Line 173-174, the four segments included in the simulations ended at residue 2190. So the pore region was not included? It does not seem to be the case elsewhere.

4. Line 270, space is missing at the beginning of this paragraph.
5. Is the in-house MATLAB script accessible?
6. Ref. 26 and ref. 41 are the same reference, just listed with different publication years (one for online and one for printed).
7. Figure 4f is a bit confusing. What are the 0 points for twist and tilt? And it is not obvious where the time points 6.3 to 6.9 μ s come from. The reader would need to dig through the Methods and do some calculation. It would be helpful to include more information in the legend.

Reviewer #3:

Remarks to the Author:

In this study, the authors explore the mechanism of action of Yoda1 on the mechanosensitive channel Piezo1.

Yoda1 is known to be an agonist-like molecule lowering the pressure sensitivity of Piezo1 channels. A recent study showed that binding of one Yoda1 molecule per channel is sufficient to induce maximal effect on Piezo1 pressure sensitivity (Lacroix et al., 2018). By using molecular dynamic simulation, site directed mutagenesis and mostly calcium imaging, the authors identify the binding pocket of Yoda1, localized on each of the three peripheral blades of the channel. Identification of Yoda1 binding site represents the major finding of this study. Moreover, simulations indicate that bound Yoda1 increases tension induced protein motion, suggesting that its binding facilitates Piezo1 force-induced conformational changes leading to channel pore opening.

The work performed here is of interest, but could be strengthened by a better characterization of binding-pocket mutants. I have two comments:

-Since calcium imaging is an indirect way to visualize ion channel activity, authors should implement their study with electrophysiological characterization of Yoda1-binding pocket mutants. According to calcium imaging experiments, A2094 mutations impair Yoda1-induced calcium signal proportionally to the volume of the amino acid replacing the alanine at this position. Authors should characterize and compare the effect of Yoda1 on pressure-sensitivity of WT, A2094V and A2094W channels using patch-clamp recordings.

-A recent study showed that L1342/L1345 Piezo1 mutations abolish Yoda1 sensitivity, without affecting its binding. Interestingly, these mutants are still activated by membrane stretch, suggesting that they affect more specifically Yoda1 modulation of Piezo1 (Wang et al., 2018). Authors should discuss this effect in light of their results.

Responses to reviewers

Reviewer #1 (Remarks to the Author):

The paper is about the determination of a possible mechanism of activation of mechanosensitive ion channel Piezo1 by agonist Yoda1. The main investigation method is based on microsecond-long all atoms molecular dynamics (MD) simulations, with additional evidences coming from calcium imaging spectroscopy and electrophysiological techniques. MD simulations of the system (Piezo1 + membrane + solvent) were carried out in different conditions (with and without tension, with and without ligands etc.)’.

The authors conclude that Yoda1 binds to a specific hydrophobic pocket sandwiched between Piezo1 Repeats A and B” and that “binding may alter a relative conformational motion between the pore region (that includes the most proximal repeat A) and the remaining N terminal part of the arm.” This suggested outcome is intriguing, and the authors support their conclusion performing a community analysis on a simplified system - with all ligands, except L13, removed - putting in evidence that L13 acts as a hinge between the two repeats. The paper presents a significant collection of arguments sustaining the authors conclusions. From the technical point of view the methods employed are presented flawlessly, with even an excess of specific details for the MD part. This could be certainly helpful in case other workers would like to attempt reproducing the observed results (although the availability of the computational tool employed, ANTON, remains limited). The statistical analysis of the experimental evidences presented in the paper is, as far as I can judge, correct.

The Discussion session is sketchy and could probably be improved with additional comments. For instance, is it possible to estimate quantitatively the nature of the motions induced by tension in the presence of Yoda1. The qualitative assertion “a large tilt, or extension of the distal part of the arm” could be likely be made more quantitative (amplitude of motion and if possible estimate of time-scale).

In any case the paper presents a persuasive and convincing picture which is certainly of interest to a significant audience, especially biophysicists and computational biochemists. I recommend publication in Nature Communications.

We thank the reviewer for this comment. We have considerably expanded our discussion and described protein motions in a more quantitative manner with a new figure clearly showing the timescale of our PCA analysis.

Reviewer #2 (Remarks to the Author):

Using molecular dynamic simulations, the authors proposed a mechanism that the small molecule Yoda1 activates mechanosensitive Piezo channel by binding to an allosteric pocket like a wedge. Although the major finding is from MD simulations, the authors did a nice job validating the proposed binding pocket, as well as excluding another possible binding site, by mutagenesis, calcium imaging, and electrophysiology. In addition, the simulations showed several features that agree well with literature: a. potassium ions accumulated near the negatively-charged fenestrations, b. the lipid bilayer forms a dome after equilibration, and c. the channel flattens with the lipid bilayer when tension is applied. This study is novel. How Yoda1 works on Piezo1, either through direct binding or via modifying the local lipid environment, is under debate in the field and would be important to help understand the mechanism of Piezo gating in general. Though interesting, there are several points that need to be addressed before publication in Nature Communications.

1. In the simulation system, 20 Yoda1 ligands were added. Based on Figure 2a and Movie S1, the 20 ligands appeared sampling the entire trimeric Piezo channel. Then why is there only one ligand binding to the L13 site in one of the three subunits (arm 3), i.e. why do other ligands not bind to the equivalent L13 sites in other subunits?

We thank reviewer for this question. The main reason why only one Yoda1 binding event is observed during the simulation is because 3-fold symmetry of the homotrimer is not imposed during the MD simulation. Therefore, the conformation of each subunit is not expected to be identical during the simulation time. Despite that simulations have reached apparent equilibrium, indicated by protein backbone RMSD began to plateau, each subunit continues to sample different conformations within the conformational ensemble during the microsecond simulations. A 100% symmetry across all subunits can only be obtained by averaging over a much longer time scale which ensure each subunit samples the full conformational space so that all the thermodynamic fluctuations can be averaged out. The asymmetry observed from our simulation is also consistent with the asymmetry observed from cryo-EM structures (Nature2018, V554, 481-486 and 487-492).

Lacroix et al. (ref. 30) suggested that only one Yoda1-binding subunit is sufficient for Yoda1-mediated activation, but it is not clear whether the activation is through a concerted or sequential mechanism. So it would be interesting to see the simulations with 3 Yoda1 ligands bound to the L13 sites of all three subunits. Maybe only then would the pore open?

This is an interesting point. However, as mentioned above, the three subunits are not symmetrical during the simulation. We have tried to manually put Yoda1 molecules in the two other binding sites but found that the other two Yoda1 bindings are not stable, at least during 100 nanosecond simulation run. In order to keep 3 Yoda1 bound to all three L13 sites, additional restraints are needed between ligand and protein, which would complicate the interpretation of our current results. We do not rule out the possibility that all three L13 site can be occupied by Yoda1, but it was not observed from our simulation and we did not intend to restraint them at the binding sites.

2. The two Piezo paralogs have high sequence identity, yet Piezo2 is insensitive to Yoda1. Could it be explained by the mechanism presented here? Although the structure of Piezo2 is not available, sequence alignment and topology prediction could be well done with reference to Piezo1. Many regions are highly conserved between Piezo1 and Piezo2, including the helices containing A1718, A2091, and A2094 (corresponding residues A, G, and A in Piezo2). So it appears that the Yoda1 binding pocket might also be present in Piezo2. Then what would be the mechanism of the insensitivity of Piezo2 to Yoda1?

This is a very interesting point. We do not know the mechanism underlying Yoda1 selectivity. Many residues in the binding site are conserved. Yet, amino-acid conservation of a homologous binding site may not retain structural features needed for inserting the small molecule in the correct pose. Hence, differences in backbone conformation or side chain orientation may prevent Yoda1 to interact with the homologous binding site in Piezo2 or to prevent Yoda1 to bind this site with an effective binding pose. On the other hand, Yoda1 may bind to the homologous binding site in Piezo2 in the correct pose, but the binding may not produce any modulatory effect, perhaps due to a loss of allosteric coupling between the binding site and the channel pore. We do not have yet enough evidence to favor one scenario or another to explain the Piezo1 vs. Piezo2 selectivity of Yoda1.

3. Due to the technical limitation of the supercomputer, which is understandable, the three peripheral repeats D-F were removed from the simulations. However, including these three repeats, which contribute significantly to the overall curvature of the channel, would likely drive the simulation more to an open state. The authors did mention that full-length arms might be required to open the pore in the simulations. But not only the structurally unresolved ones, but these three structurally available repeats could also well be essential. It is important to state that.

We agree with the review's comment that the three peripheral repeats D-F are likely important for mechanosensitivity of Piezo1. Previous studies have shown that deleting the extracellular loops that connect TM15-16 of repeat F and TM19-20 of repeat E retain normal Yoda1 sensitivity but renders Piezo1 insensitive to stretch and less sensitive to poking (Zhao et al., Nature 2018).

4. The authors chose POPC as the lipid component of the membrane without giving any reason. This is a commonly used lipid, but Piezo and Yoda1 may actually require an optimized set of lipids to work properly. Bending moduli, intrinsic curvature and symmetry of the lipid bilayer could all be critical to open this unusually curved channel. The authors may take advantages of simulations to explore different lipid compositions in Piezo gating. The actual work may be well beyond the scope of this manuscript, but it is worth some solid discussion here.

The reviewer raised an important point. Indeed, Piezo1 channel is tightly modulated by lipid component such as phosphatidylserine and phosphoinositide. Experiments done in the Patapoutian group however show that Yoda1 can open Piezo1 in a symmetrical artificial bilayer, although in these studies the membrane contains DphPC (which is often preferred due to its improved chemical stability over other lipids such as POPC).

The nature of the modulation of Piezo1 by negatively-charged lipids remains unclear, and as the reviewer pointed out, MD simulation is a useful tool to study Piezo1 in different lipid environments. However, to pinpoint the interaction between Yoda1 and Piezo1, it is important to keep the rest of the system as simple as possible. If we start a Yoda1 binding simulation with an inhomogeneous membrane composition, it will be difficult to discriminate ligand-induced from lipid-induced protein conformational changes. Ideally, one would like to simulate any membrane protein in a realistic plasma membrane component, however, not all the lipid force fields are well parameterized and the timescale for equilibrium in an inhomogeneous membrane is also longer. This will introduce additional uncertainty and difficulty in interpreting our simulation results. Since this work represents the first (to our knowledge) all-atom MD simulation of Piezo1, we choose to use POPC, which is the golden standard lipid model for current membrane protein simulations. This system will also serve as a control system to compare with future Piezo1 simulations in other lipid components.

5. Related to the effect of lipid on Piezo, during the simulations, the authors carefully set the restraints so that Yoda1 molecules are not far in the surrounding membrane environment. What if Yoda1 could also work by modulating membrane property since it readily diffuses into the lipid? Could this be captured by the simulations? Or have the authors already tested that and excluded this possibility? It would be important to know if this is the case.

The restraints are quite large (45 angstroms), so the ligands can diffuse freely in the membrane bilayer around each channel subunit. This is a more efficient sampling than simply allowing the ligands to diffuse throughout the whole simulation box. Having said that, contributions from Yoda1-induced lipid modulation is unlikely because of the two following experimental evidences:

1) the effect of Yoda1 is abolished by point mutations and chimeras (e.g. A2094W from this work, L1342A/L1345A from Wang et al. 2018 and the Piezo1/Piezo2 1961-2063 chimera from Lacroix et al. 2018);

2) Many Yoda1 analogs that have nearly identical physico-chemical properties as Yoda1 (e.g. Chloro- to Fluoro- substitutions) do not activate Piezo1 (Syeda et al. 2015 and Evans et al. 2018).

6. The authors presented PCA analysis of the subset corresponding to the ten 30 ns trajectories over which membrane tension increased from 20 to 40 mN/m (time points from 6.3 to 6.9 μ s in Figure 4f, right?) because this subset yielded the largest global changes in structure. Since other subsets were also subjected to the same analysis, did they give consistent, just smaller, changes? Or did they show different patterns? It would be interesting to know if Piezo moves differently under different tension regimes. Especially if the authors would like to claim the twist-tilt-twist motion sequence, the two-dimensional projection analysis should include earlier time points since the tilt angle started to increase between 5.4 to 6.0 μ s according to Figure 4d.

We thank reviewer for this question. The PCA analysis now uses the whole membrane stretching trajectory from 0 to 40 mN/m (time points 4.8 to 7.9 μ s). It is consistent with the principal motions captured using the 6.3 to 6.9 μ s. More specifically, the tilt and twist motions are the two dominant motions of Piezo1 during membrane stretching. To clarify this point, we

have replaced Figure 4f with the PCA projection of the whole stretching trajectory. It is clear that the protein first undergoes arm twist between 4.8~5.8 μ s (show in blue dots), and then undergo a series of tilt motion accompanied by small scale twist motion between 5.8~7.2 μ s (show from cyan to orange dots), and twist motion at the end (show in red dots). Therefore, our previous plot represents a subset of the same motion.

Minor points:

1. This manuscript only discussed Yoda1, not other Piezo1 chemical activators such as Jedi1 and Jedi2, which work through the extracellular side, thus likely a different mechanism compared to Yoda1. Therefore, the title might be too broad which implies a general mechanism for all Piezo1 chemical activators.

This is true, we have changed our title accordingly.

2. Line 80, the mutant A1718A should be A1718W.

Corrected, thank you!

3. Line 173-174, the four segments included in the simulations ended at residue 2190. So the pore region was not included? It does not seem to be the case elsewhere.

That was mistake, thank you for spotting it.

4. Line 270, space is missing at the beginning of this paragraph.

Corrected, thank you!

5. Is the in-house MATLAB script accessible?

Yes, we have supplied our script with our revisions

6. Ref. 26 and ref. 41 are the same reference, just listed with different publication years (one for online and one for printed).

Corrected, thank you!

7. Figure 4f is a bit confusing. What are the 0 points for twist and tilt? And it is not obvious where the time points 6.3 to 6.9 μ s come from. The reader would need to dig through the Methods and do some calculation. It would be helpful to include more information in the legend.

We have modified this figure and included a timescale on the legend.

Reviewer #3 (Remarks to the Author):

In this study, the authors explore the mechanism of action of Yoda1 on the mechanosensitive channel Piezo1.

Yoda1 is known to be an agonist-like molecule lowering the pressure sensitivity of Piezo1 channels. A recent study showed that binding of one Yoda1 molecule per channel is sufficient to induce maximal effect on Piezo1 pressure sensitivity (Lacroix et al., 2018). By using molecular dynamic simulation, site directed mutagenesis and mostly calcium imaging, the authors identify the binding pocket of Yoda1, localized on each of the three peripheral blades of the channel. Identification of Yoda1 binding site represents the major finding of this study. Moreover, simulations indicate that bound Yoda1 increases tension induced protein motion, suggesting that its binding facilitates Piezo1 force-induced conformational changes leading to channel pore opening.

The work performed here is of interest, but could be strengthened by a better characterization of binding-pocket mutants. I have two comments:

-Since calcium imaging is an indirect way to visualize ion channel activity, authors should implement their study with electrophysiological characterization of Yoda1-binding pocket mutants. According to calcium imaging experiments, A2094 mutations impair Yoda1-induced calcium signal proportionally to the volume of the amino acid replacing the alanine at this position. Authors should characterize and compare the effect of Yoda1 on pressure-sensitivity of WT, A2094V and A2094W channels using patch-clamp recordings.

Patch-clamp recordings for the A2094W mutant were previously included and we are now including examples of WT traces for comparison. Several attempts to record from the A1718W mutant were unsuccessful, suggesting this mutant fails to respond to membrane stretch. Hence, it is quite possible that our observed correlation between the amplitude of Yoda1-induced calcium entry and the size of the side chain at position A1718 may be contributed by intrinsic alterations of channel function induced by A1718 mutations. We have cautiously removed this interpretation from our paper.

To further strengthen our results, we also report the response of several mutations at positions A2094 and A1718 to Jedi2 using calcium imaging. Remarkably, the A2094W mutant, which remains stretch-sensitive, is not activated by Jedi2 whereas the A1718W mutant, which failed to respond to membrane stretch, retains some sensitivity to Jedi2. Since the R2135A mutant is not activatable by stretch (Coste et al., 2015), we used R2135A as a negative control in these experiments. We found that this mutant responds robustly to both agonists with amplitude similar to WT channels. These results show that mechanical and chemical activation pathways are somehow relatively independent in Piezo1 and that some mutations may affect one pathway without affecting the other.

-A recent study showed that L1342/L1345 Piezo1 mutations abolish Yoda1 sensitivity, without affecting its binding. Interestingly, these mutants are still activated by membrane stretch, suggesting that they affect more specifically Yoda1 modulation of Piezo1 (Wang et al., 2018). Authors should discuss this effect in light of their results.

Indeed, the phenotype of the L1342A/L1345A (LL/AA) mutant is quite interesting. In addition to losing its sensitivity to Yoda1, the LL/AA mutant is also not sensitized by Jedi1 (Fig 6 in Wang et

al. 2018), suggesting that the loss of chemical activation is not specific to Yoda1 but extends to at least another small molecule agonist acting from a different protein region than Yoda1 (Wang et al. 2018). In addition, while the LL/AA mutant remains stretch-sensitive, its response to stretch is quite diminished compared to WT Piezo1, with a ~20 mm Hg negative shift of the P50 values (Fig 6 in Zhao et al. 2018). The LL/AA mutation also lost its activation by cell poking (Fig 6 in Zhao et al. 2018). Hence, the LL/AA mutations seem to alter at least some of its mechanotransduction machinery.

Since the LL/AA mutant binds Yoda1, the loss of agonist sensitivity likely arises from an alteration of the allosteric coupling between the agonist binding site and the channel pore. Mutations that abolish this chemical transduction pathway may also alter to some degree the channel's mechanical transduction pathway(s) and *vice versa*. This interplay between chemical and mechanical transduction pathways could explain why some Yoda1-insensitive mutants are mechanosensitive or partially mechanosensitive (LL/AA mutant, A2094W, chimera 1961-2063), while some mechano-insensitive mutants remain chemically activated by Yoda1 and Jedi2 (R2135A). Future structure-functions studies will be needed to better understand the structural bases for these remarkable mutant phenotypes.

Reviewers' Comments:

Reviewer #2:

Remarks to the Author:

I appreciate the authors' considerable effort in addressing my concerns. I believe the manuscript is now improved considerably. No further comments.